# Association of Nonalcoholic Fatty Liver Disease (NAFLD) with Peripheral Diabetic Polyneuropathy: A Systematic Review and Meta-Analysis

**DOI:** 10.3390/jcm10194466

**Published:** 2021-09-28

**Authors:** Carla Greco, Fabio Nascimbeni, Francesca Carubbi, Pietro Andreone, Manuela Simoni, Daniele Santi

**Affiliations:** 1Unit of Endocrinology, Department of Biomedical, Metabolic and Neural Sciences, University of Modena and Reggio Emilia, 42121 Modena, Italy; manuela.simoni@unimore.it (M.S.); daniele.santi@unimore.it (D.S.); 2Unit of Endocrinology, Department of Medical Specialties, Azienda Ospedaliero-Universitaria of Modena, Ospedale Civile di Baggiovara, 41125 Modena, Italy; 3Division of Internal Medicine and Metabolism, Department of Internal Medicine, Azienda Ospedaliero-Universitaria of Modena, Ospedale Civile di Baggiovara, 41125 Modena, Italy; fabio.nascimbeni@libero.it (F.N.); francesca.carubbi@unimore.it (F.C.); pietro.andreone@unimore.it (P.A.); 4Unit of Internal and Metabolic Medicine, Department of Biomedical, Metabolic and Neural Sciences, University of Modena and Reggio Emilia, 42121 Modena, Italy

**Keywords:** NAFLD, diabetes mellitus, peripheral polyneuropathy

## Abstract

Aims. The relationship between nonalcoholic fatty liver disease (NAFLD) and diabetic polyneuropathy (DPN) has been demonstrated in many studies, although results were conflicting. This meta-analysis aims to summarize available data and to estimate the DPN risk among NAFLD patients. Materials and methods. We performed a comprehensive literature review until 4 June 2021. Clinical trials analyzing the association between NAFLD and DPN were included. Results. Thirteen studies (9614 participants) were included. DPN prevalence was significantly higher in patients with NALFD, compared to patients without NAFLD (OR (95%CI) 2.48 (1.42–4.34), *p* = 0.001; I2 96%). This finding was confirmed in type 2 diabetes (OR (95%CI) 2.51 (1.33–4.74), *p* = 0.005; I2 97%), but not in type 1 diabetes (OR (95%CI) 2.44 (0.85–6.99), *p* = 0.100; I2 77%). Also, body mass index and diabetes duration were higher in NAFLD subjects compared to those without NAFLD (*p* < 0.001), considering both type 2 and type 1 diabetes. Conclusion. Despite a high heterogeneity among studies, a significantly increased DPN prevalence among type 2 diabetes subjects with NAFLD was observed. This result was not found in type 1 diabetes, probably due to the longer duration of disease. Physicians should pay more attention to the early detection of DPN, especially in patients with NAFLD.

## 1. Introduction

Peripheral diabetic polyneuropathy (DPN) is a microvascular complication of diabetes mellitus (DM), representing the most clinically relevant manifestation of typical forms of diabetic neuropathy (DN). DPN has been defined by the Toronto Expert Panel on Diabetic Neuropathy as a symmetrical, length-dependent sensorimotor polyneuropathy attributable to metabolic and microvascular alterations, resulting from the chronic hyperglycemia typical of diabetes and cardiovascular risk covariates [1]. DPN occurs in at least 20% of people with type 1 DM (T1DM) after 20 years of disease duration, as suggested by large observational cohorts [2,3] and the Diabetes Control and Complications Trial (DCCT)/Epidemiology of Diabetes Interventions and Complications (EDIC) Study [4,5]. Considering type 2 DM (T2DM), DPN has been detected in at least 10–15% of newly diagnosed patients with T2DM [6,7], and up to 50% after 10 years of disease duration [8,9]. Moreover, DPN has been identified in 11% to 23% of people with prediabetes [10]. From a clinical point of view, the DPN diagnosis is extremely relevant in DM management, since it confers a predisposition to pain, numbness, ulceration, and amputation of the distal extremities, increasing the risk of all-cause and cardiovascular disease mortality [11,12,13].

Starting from the clinical relevance of DPN, many authors tried to identify factors able to predict DPN in DM. Till now, diabetes duration and glycemic control, expressed by glycated hemoglobin (HbA1c), represent the main predictive factors [14]. Moreover, metabolic syndrome components, such as hypertriglyceridemia, hypertension, abdominal obesity, and low high-density lipoprotein (HDL) serum levels, are consistently associated with DPN in both T2DM and T1DM [15,16]. Alongside metabolic variables, several lifestyle habits have been detected as further correlates, such as smoking, alcohol abuse, height, and older age [15]. In general, many studies suggested that DPN prevalence was higher in cases of concomitant comorbid conditions, such as micro- (nephropathy or retinopathy), macro- vascular disease (peripheral arterial disease or cardiovascular disease) and depression [17]. Finally, new biochemical markers have been investigated as potential predictive markers of DPN. In particular, novel systemic biomarkers of oxidative stress (i.e., reactive oxygen species), inflammation (interleukin (IL)-6, and tumor necrosis factor (TNF)-a), and vascular activation, have been linked to distal DPN development [16].

Recently, DPN has been associated with another pathological condition linked to DM and obesity, the nonalcoholic fatty liver disease (NAFLD). NAFLD is a metabolically derangement-based liver disease, defined by the presence of steatosis in more than 5% of hepatocytes, in association with metabolic risk factors (such as obesity, diabetes, and dyslipidemia) and in the absence of excessive alcohol consumption or other chronic liver diseases [18,19]. NAFLD affects more than 25% of the global population [20] and is largely demonstrated as highly prevalent in patients with T2DM (60–75%) [21]. Many studies proved that NAFLD is associated with an increased risk of macro- and micro-vascular complications in diabetic patients [22,23,24], notably including albuminuria [25] and retinopathy [26]. Currently, there is little information about the association between NAFLD and DPN, and the available data are scarce and conflicting.

With this in mind, this meta-analysis was conducted to summarize available data estimating the DPN prevalence among diabetic patients with NAFLD. In particular, the study was designed to highlight potential links between NAFLD and DPN in diabetic patients.

## 2. Materials and Methods

This meta-analysis was performed according to the Cochrane Collaboration and PRISMA statement. To ensure originality and transparency of the review process, the meta-analysis was registered in the International Prospective Register of Systematic Reviews (PROSPERO; registration ID 251792).

The literature search was performed until 4 June 2021 considering the following string: ((((((((((diabetes) OR (type 2 diabetes mellitus)) OR (type 1 diabetes mellitus)) OR (diabetes mellitus)) OR (T2DM)) OR (T1DM)) AND (neuropathy)) OR (peripheral neuropathy)) AND (NAFLD)) OR (non-alcoholic fatty liver disease)) OR (hepatic steatosis). Medline, Embase, and Cochrane databases were considered.

Since the term NAFLD was coined in 1980 to describe fatty liver disease arising in the absence of significant alcohol intake [27], studies published before 1980 were excluded from the analysis. Moreover, since the NAFLD diagnosis could be achieved by different methodologies, we considered studies in which the presence of NAFLD was evaluated either by liver ultrasound, composite non-invasive biomarkers, or ultrasound elastography.

### 2.1. Endpoints

Primary endpoint was the prevalence of peripheral DPN evaluated by either, signs, symptoms, or nerve conduction study (NCS). All variables known to predict DPN were considered as secondary endpoints. In particular, risk factors and clinical correlates of DPN are age, diabetes duration, glycaemic control, arterial hypertension, and smoking [14,28,29]. Thus, the following variables were extracted from the included studies as secondary endpoints: patient’s age, height, body mass index (BMI), HbA1c, HDL, triglycerides and total cholesterol, C-peptide serum levels, and diabetes duration. Lipid profile-related variables were transformed in mmol/L when reported differently in the original works. Moreover, smoking and alcohol habits were extracted when available.

### 2.2. Study Selection and Inclusion Criteria

The literature search evaluated all clinical trials with the following inclusion criteria: (i) either interventional or observational study designs, (ii) in which the DPN prevalence was reported, (iii) in people with DM, and (iv) in which baseline presence or absence of NAFLD was assessed. Both T1DM and T2DM were considered eligible. No specific exclusion criteria have been considered for the studies extracted. Moreover, both longitudinal and cross-sectional studies were included in the analysis. Since the analysis was not focused on a specific pharmacological or not-pharmacological intervention, the randomization was not considered as inclusion criterion.

### 2.3. Data Collection Process and Quality

Two authors (C.G. and D.S.) separately performed the literature search, collecting abstracts of each study. Each abstract was evaluated for inclusion criteria and data were extracted from each study considered eligible. C.G. and D.S. performed quality control checks on extracted data. For the literature search, the primary endpoint was the prevalence of DPN, thus, patients were divided in study and control groups considering the presence or absence of NAFLD, respectively.

The two investigators extracted the following information from the included studies: (i) general characteristics including study design, sample size, and year of publication, (ii) diagnostic methods of NAFLD and of DPN, (iii) the proportion of NAFLD patients and diabetic neuropathy patients, and (iv) adjusted confounders.

All variables were extracted as mean ± standard difference. When variables were reported as median (interquartile or minimum and maximum) in the original work, they were transformed accordingly.

### 2.4. Data Synthesis and Analysis

The analyses were performed using the Review Manager (RevMan) 5.4 Software (Version 5.4.1, The Nordic Cochrane Centre, The Cochrane Collaboration; Copenhagen, Denmark, 2014). Values of *p* < 0.05 were considered statistically significant.

The DPN prevalence was compared between diabetic patients with and without NAFLD, considering the odds ratio obtained applying the Mantel-Haenszel method. The fixed model was initially used, whereas the random effect model was applied in case of I^2^ higher than 60%. The heterogeneity degree among different studies was examined by inspecting both the scatter in the data points and the overlap in their confidence intervals (CIs), and by performing I^2^ statistics. Weighted mean differences and 95% CIs were estimated for the literature search. Continuous data were expressed as mean differences when reliable methods have been used for detection (i.e., BMI, HbA1c, etc.), whereas standard mean differences were used for other variables (i.e., lipid asset).

Sensitivity analyses were performed in order to reduce the studies’ heterogeneity. In particular, patients’ inclusion criteria were evaluated and used to divide the included studies. Moreover, sensitivity analyses were performed considering study design, dividing longitudinal from cross-sectional studies. Finally, publication biases were explored through funnel plots [30] and corrected by Duval and Tweedie’s ‘trim-and-fill’ analysis [31]. In the presence of asymmetric funnel shapes, this test detects putative missing studies to rebalance the distribution and provides an adjusted pooled estimate taking the additional studies into account, thus correcting the analysis for publication bias.

## 3. Results

The literature search identified 2613 papers, after duplication removal. After abstract evaluations, 16 studies were considered for the full text analysis [32,33,34,35,36,37,38,39,40,41,42,43,44,45,46,47] (Figure 1). Three studies have been excluded, since one is a duplication of previously published results [45], and two works did not report the prevalence of neuropathy in NAFLD positive and negative patients [43,44] (Figure 1). Finally, 13 cross-sectional studies were included. No longitudinal trials were available on the topic. Table 1 summarizes the characteristics of 13 studies, finally included in the meta-analysis.

A total of 9614 diabetic patients were included in the analysis. The majority of studies (77%) evaluated T2DM patients and only three studies (23%) enrolled subjects with T1DM. All included studies enrolled both males and females, thus the role of gender could be not ruled out.

DPN prevalence was significantly higher in diabetic patients with NALFD, compared to patients without NAFLD (*p* = 0.001) (Figure 2). In particular, the prevalence of DPN was significantly higher in patients with NAFLD compared to patients without it, considering T2DM (*p* = 0.005), but not T1DM (*p* = 0.100) (Figure 2).

### 3.1. Secondary Endpoints

Secondary endpoints were considered in patients with (study group) and without (control group) NAFLD in order to evaluate which variables could affect the association with DPN. However, despite the large literature supporting the predictive role of several parameters on NAFLD, not all studies included in our meta-analysis reported these endpoints. Indeed, C-peptide, for example, was reported only in two included studies [34,40] and it could not be meta-analyzed. Similarly, smoking and alcohol habits were reported in a limited number of studies, not allowing a meta-analytic comprehensive evaluation.

In this setting, no differences in the age of patients in the study compared to control groups were observed (mean difference −0.3: 95%CI −1.9, 1.4 years, *p* = 0.720). The comparison between study and control groups in males and females separately was not performed, since the DPN percentage was not reported in each gender separately.

On the contrary, anthropometric variables demonstrated a potential predictive role. Indeed, BMI, reported in 11 included studies, was higher in the studies compared to control groups (*p* < 0.001), considering both T2DM (*p* < 0.001) and T1DM (*p* = 0.030) (Figure 3).

Considering diabetes duration and control, patients with NAFLD demonstrated no different HbA1c serum levels (*p* = 0.060) compared to controls (mean difference: 1.25, 95%CI: −0.07, 2.57), neither in T2DM (mean difference: 0.29, 95%CI: −0.58, 1.16, *p* = 0.510) nor in T1DM (mean difference: 6.49, 95%CI: −14.09, 27.07, *p* = 0.540); HbA1c serum levels were different between study and control groups. On the contrary, diabetes duration was significantly higher in the study than in control groups (*p* < 0.001), also considering T2DM (*p* = 0.006) and T1DM (*p* = 0.030) alone (Figure 4).

Considering the lipid profile, total (*p* = 0.250) (Figure 5), HDL (standard mean difference −0.13: 95%CI: −0.26, 0.01 mmol/L, *p* = 0.060), and cholesterol and triglycerides (*p* = 0.050) (Figure 6) did not differ between study and control groups. However, sensitivity analyses demonstrated higher total (*p* = 0.010) cholesterol levels in patients with NAFLD and T2DM, but not in T1DM (Figure 5). On the contrary, HDL cholesterol did not differ between the study and control groups in T2DM (standard mean difference −0.11: 95%CI: −0.25, 0.03 mmol/L, *p* = 0.130) and T1DM (−0.24: 95%CI: −0.73, 0.25 mmol/L, *p* = 0.330), separately. Finally, triglycerides were significantly higher in the study groups compared to the controls in T2DM (*p* < 0.001) and lower in T1DM (*p* = 0.009) (Figure 6).

### 3.2. Sensitivity Analyses

In order to reduce the high heterogeneity among studies (I^2^ = 97%, Figure 2), sensitivity analyses were conducted selecting those patients’ criteria that were significantly different between the study and control groups. Thus, sensitivity analyses were performed considering the patients’ BMI and diabetes duration. Moreover, since the diagnostic methods used to define DPN and NAFLD were variable between studies, further sensitivity analyses were performed considering these factors.

First, studies were divided in three groups according to patients’ BMI: normal weight (mean BMI < 25 kg/m^2^; number of studies: 4), overweight (BMI between 25 and 30 kg/m^2^; number of studies: 2), and obesity (BMI > 30 kg/m^2^; number of studies: 3). The studies’ heterogeneity remained high in three groups (92, 94, and 97%, respectively). Moreover, the limited number of studies in each subgroup reduced the statistical significance of the association between DPN and NAFLD (BMI < 25 kg/m^2^ OR (95% CI) 1.36 (0.69–2.69), *p* = 0.380; BMI between 25 and 30 kg/m^2^ OR (95% CI) 0.97 (0.28–3.40), *p* = 0.960; BMI > 30 kg/m^2^ OR (95% CI) 4.58 (0.24–5.91), *p* = 0.310). Second, studies were divided according to diabetes duration in “recent diagnosis”, when DM lasted less than 5 years (number of studies: 7) and “long-term diagnosis”, when the diagnosis dated more than 5 years before (number of studies: 6). DPN prevalence remained significantly higher in diabetic patients with NAFLD and longer DM diagnosis (OR (95% CI) 3.13 (1.32–8.70), *p* = 0.009; I^2^ 95%), but not in those with shorter diagnosis (OR (95% CI) 0.99 (0.18–3.67), *p* = 0.980; I^2^ 97%). Third, studies were divided in two groups, according to the diagnostic criteria of DPN. In particular, probable neuropathy was defined when only signs and symptoms have been considered (number of studies: 7), whereas confirmed neuropathy was defined by NCS (number of studies: 4). However, also this subdivision did not reduce the heterogeneity among studies (probable DPN: OR (95% CI) 3.41 (1.28–9.05), *p* = 0.010; I^2^ 97%; confirmed DPN OR (95% CI) 2.44 (1.19–3.12), *p* = 0.020; I^2^ 91%). Finally, studies were divided according to the method applied for NAFLD diagnosis in ultrasound-based diagnosis (number of studies: 10) and Fibroscan-based diagnosis (number of studies: 3). DPN prevalence remained significantly higher in diabetic patients with NALFD detected using ultrasound methods (OR (95% CI) 2.43 (1.21–4.89), *p* = 0.002; I^2^ 97%), but not using Fibroscan (OR (95% CI) 0.98 (0.30–3.23), *p* = 0.970; I^2^ 97%). This latter finding, however, seems to be due to the low number of studies using this methodology.

### 3.3. Publication Bias

Although the asymmetric shape of the funnel plot suggested a possible publication bias (Figure 7), in particular regarding studies enrolling T2DM, the trim-and-fill analysis did not identify putative missing studies.

## 4. Discussion

Several studies have been designed to explore the impact of NAFLD on DPN prevalence in both T1DM and T2DM patients so far, but this is the first attempt to systematically combine these results together in a meta-analysis. We demonstrate that DPN is more frequent when NAFLD is associated to DM, evaluating more than 9000 diabetic subjects. This result has an immediate clinical translation. Indeed, we clearly demonstrate that a diabetic patient must be carefully evaluated for the onset of peripheral neurological complications, especially when NAFLD is associated with diabetes. This is particularly true in T2DM or in T1DM and advancing age. Indeed, we demonstrate that DPN risk in T1DM is higher when the diabetes duration is longer, confirming that the long disease duration could be a confounding factor for DPN development. Moreover, here we highlight how NAFLD in DM is strictly related to high BMI and diabetes duration, confirming how the prevention of the DM complications must necessarily involve attention to weight gain. In details, NAFLD determines a complex array of metabolic and extra-hepatic consequences, which result from the intra-hepatic deposition of ectopic fat. This condition strongly correlates with abdominal obesity, insulin resistance, and all components of metabolic syndrome. Notably, obesity is one of the clinical correlates of PND in DM2 people. Therefore, obesity itself could represent a confounding factor of the association between NAFLD and PND, at least in DM2.

The link between NAFLD and microvascular complications in diabetic patients is based so far only on association studies, but the cause-effect relationship is far from being completely elucidated. In particular, NAFLD has been suggested as an independent predictor for diabetic kidney disease and proliferative diabetic retinopathy in patients with T2DM [48,49], while the association with DPN is more debated. Thus, NAFLD has been considered as a risk factor for organ-specific complications of DM. What is largely supposed is that NAFLD could exacerbate insulin resistance, impairs dyslipidemia, and predisposes vessels to atherogenic damages, throughout the release of pro-inflammatory, pro-coagulant, and pro-atherogenic factors [50,51,52]. Moreover, NAFLD induces those damages leading to endothelial dysfunction, predisposing to vascular diseases [53].

In particular, considering DPN, the pathogenetic relationship with NAFLD is still under debate. From one side, the metabolic asset leading to NAFLD is largely considered among the risk factors for DPN development. Moreover, in addition to the known metabolic correlates, the possible molecular mediators linking NAFLD with DPN could include the increased release of some pathogenic mediators from the liver, such as advanced glycation end-products, reactive oxygen species, C reactive protein, IL-6, and TNF-α, as also suggested for retinopathy and chronic kidney disease [49].

NAFLD natural history describes early stages, typically asymptomatic, with only incidental finding of abnormal liver enzymes, such as raised plasma alanine aminotransferase (ALT), aspartate transaminase (AST), and/or gamma-glutamyltransferase (γGT) [54]. However, since liver enzymes largely fluctuate in NAFLD patients, they are not routinely considered as clinical markers of NAFLD diagnosis or severity [18,55,56]. Thus, the use of imaging techniques, such as ultrasound, is generally applied as a first line diagnostic step in evaluating hepatic steatosis, also considering its safety and availability, and low cost [57]. With this in mind and considering the main result of our meta-analysis, it is clear that diabetic patients must undergo hepatic ultrasound evaluation in order to precociously detect the presence of NAFLD. However, NAFLD severity could also have a role in comorbidities development. The gold standard to detect NAFLD severity, in terms of steatosis amount, necro-inflammation, and fibrosis is represented by liver biopsy [18] that is not suitable for large-scale screening purposes, due to invasiveness and costs. Several emerging non-invasive techniques, notably including composite biomarkers, ultrasound elastography, or magnetic resonance, display good performance in evaluating NAFLD severity and have been proposed for widespread use in clinical practice. Unfortunately, only a minority of studies enrolled in our meta-analysis evaluated NAFLD severity, preventing us from a reliable analysis of such data. Future studies should be designed to identify whether NAFLD severity could predict the DPN development.

The result of this meta-analysis provides a clear snapshot on what we know about the association among DM, NAFLD, and peripheral DPN. This setting, however, is very heterogeneous. Indeed, all studies enrolled are population-based matched case-control studies, with a clear difference among inclusion and exclusion criteria. Moreover, only in 7 out of 13 studies the aim of the study was the evaluation of the prevalence of micro-vascular complications in diabetic patients. In the remaining part, the peripheral DPN has been assessed in relation to specific clinic or biochemical characteristics of enrolled patients, such as uric acid or liver fibrosis. Thus, the approach to our study question (i.e., whether NAFLD predisposes to peripheral DPN in patients with DM) is widely different, limiting the robustness of a comprehensive evaluation. Moreover, the peripheral DPN can be probable and confirmed, according to the diagnostic path followed. In particular, only when NCS is performed, a confirmed diagnosis should be reached. Our meta-analysis collected only six studies in which a confirmed DPN could be verified, increasing the heterogeneity among studies. Furthermore, the clinical management of patients enrolled in each study is extremely variable. In particular, diabetic comorbidities and complications management could have a significant role in DPN development. Here, however, we could not adjust the meta-analytic approach with the therapies applied to enrolled patients. This could lead to confounding results considering secondary endpoints. As a confirmation, we highlight that lipid profile does not change between the study and control group, although several studies suggested a worse lipid profile in patients with NAFLD [58,59,60]. Thus, we could not speculate in favour or in contrast to dyslipidaemia as a factor linking NAFLD and peripheral neuropathy in diabetic patients.

Our study presents several limitations. First, this is a meta- analysis and could not determine causal relations. Second, we found considerable heterogeneity among studies, which limits the exportability of our results. Third, the diagnostic criteria for DPN displays some methodological heterogeneity, resulting in a probable diagnosis in some studies and a confirmed diagnosis of DPN in others. Furthermore, in enrolled trials, the diagnostic method used to evaluate NAFLD was quite heterogeneous, and liver biopsies, which are the gold standard to evaluate NAFLD severity and may play a probable role in DPN onset and progression, were not performed in any study. Finally, differences in country and geographic origin among studies may be one of the sources of heterogeneity, which should be treated with caution and confirmed in further research. Despite these limitations, these results suggest, in a very large sample, that DM combined with NAFLD is positively associated with peripheral DPN.

## 5. Conclusions

The present meta-analysis suggests a significantly increased DPN prevalence among diabetic patients with NAFLD, in particular in the case of T2DM. Indeed, T2DM combined with NAFLD demonstrated a higher prevalence of peripheral DPN than the T2DM-alone group. This result has not been confirmed in T1DM, likely due to the longer duration of disease as a confounding factor. Moreover, our findings confirm that NAFLD in DM is strictly related to high BMI and also to diabetes duration. In conclusion, these results suggest that physicians should pay more attention to the early detection of DPN, especially in patients with NAFLD. Lastly, large-scale prospective studies are required to elucidate causal associations between NAFLD and the microvascular complications, including DPN, in diabetic people.

## Figures and Tables

**Figure 1 jcm-10-04466-f001:**
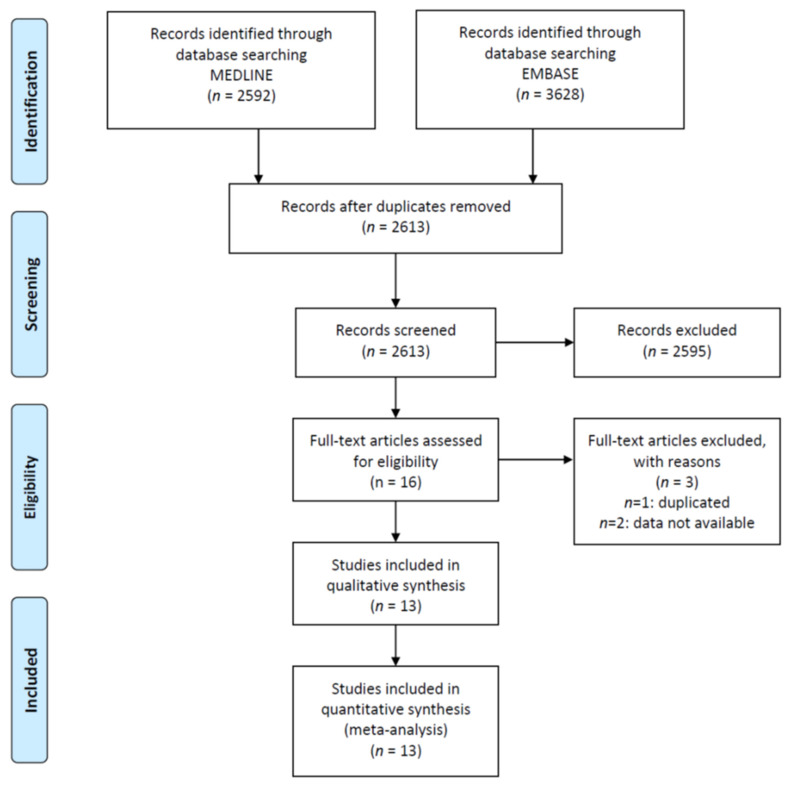
Study flow-chart.

**Figure 2 jcm-10-04466-f002:**
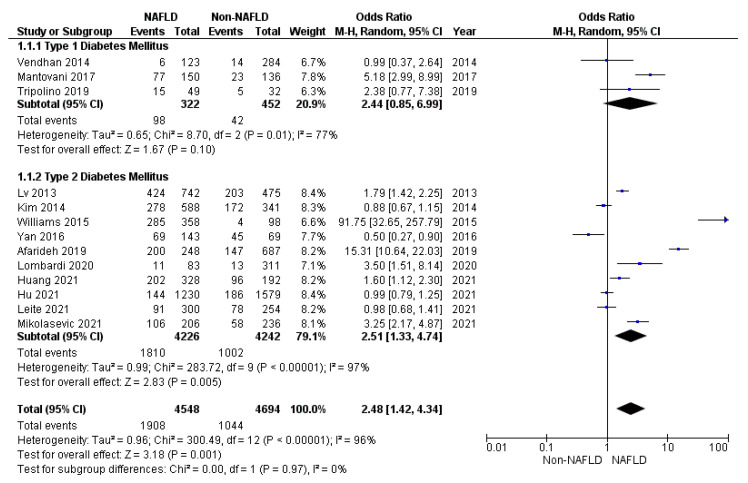
Forrest plot displaying the prevalence of peripheral neuropathy expressed as odds ratio, comparing diabetic patients with and without NAFLD.

**Figure 3 jcm-10-04466-f003:**
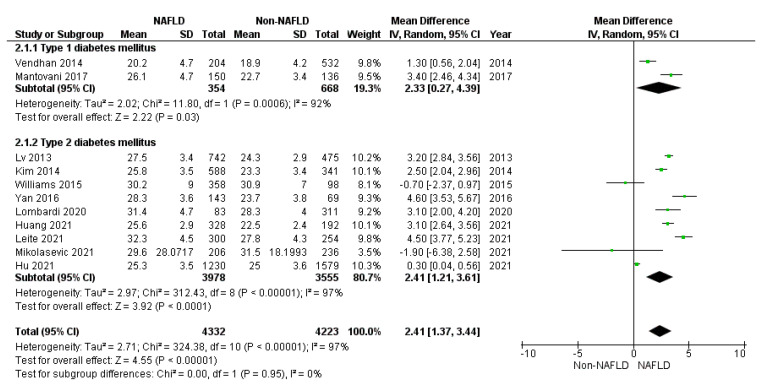
Forrest plot demonstrating the body mass index (BMI) comparing diabetic patients with and without NAFLD. Data are expressed as the mean difference and evaluated by applying the random effect model.

**Figure 4 jcm-10-04466-f004:**
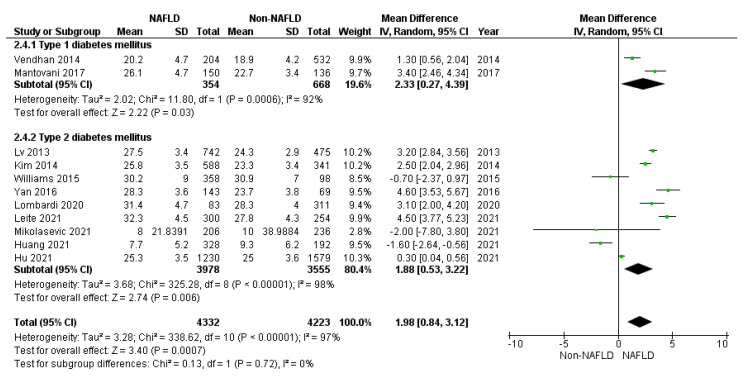
Forrest plot displaying diabetes duration in years, comparing diabetic patients with and without NAFLD. Data are expressed as a mean difference, evaluated by applying the random effect model.

**Figure 5 jcm-10-04466-f005:**
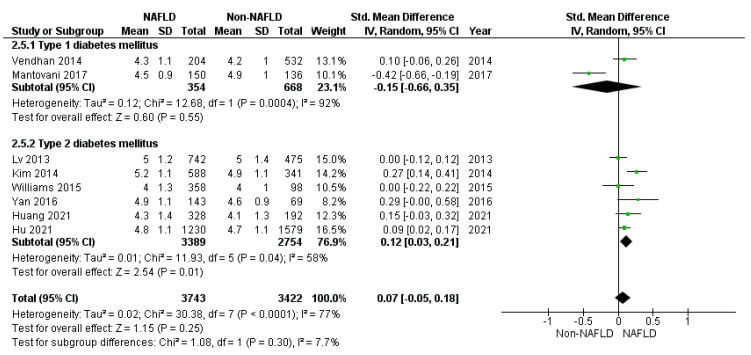
Forrest plot displaying total cholesterol serum levels expressed in mmol/L, comparing diabetic patients with and without NAFLD. Data are expressed as a standard mean difference, evaluated by applying a random effect model.

**Figure 6 jcm-10-04466-f006:**
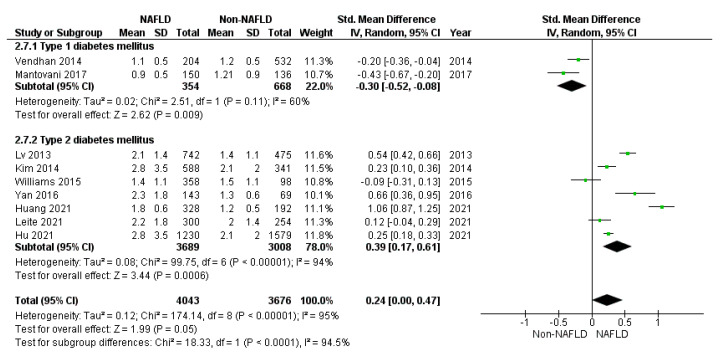
Forrest plot displaying triglycerides serum levels expressed in mmol/L, comparing diabetic patients with and without NAFLD. Data are expressed as a standard mean difference, evaluated by applying the random effect model.

**Figure 7 jcm-10-04466-f007:**
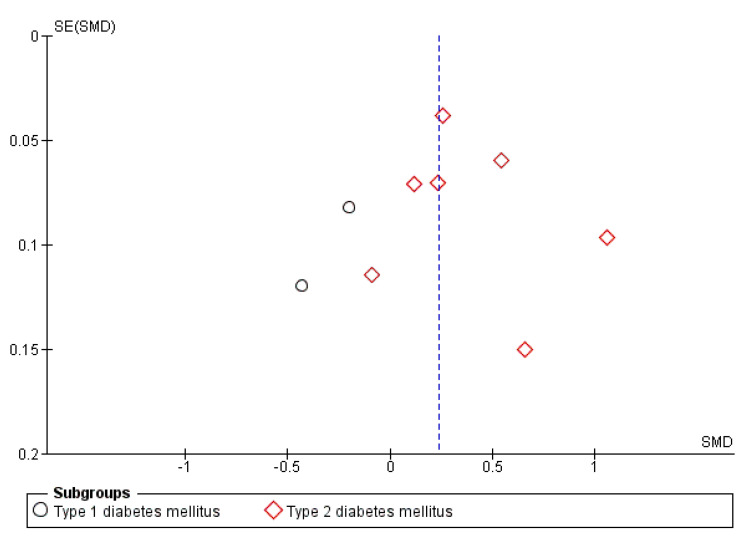
Funnel plot of the results from studies assessing the neuropathy prevalence in diabetic patients with NAFLD. SE: standard error, SMD: standardized mean difference.

**Table 1 jcm-10-04466-t001:** Included studies’ characteristics.

Author, Year	Country	Study Design	Aim of the Study	Enrolled Patients (Number) and Type of DM	Sex	NAFLD Diagnosis	DPN Diagnosis
Afarideh, 2019	Iran	Cross-sectional	To evaluate associations of serum liver enzymes and NAFLD with chronic microvascular complications in patients with T2D	935 New-onset T2DM	Males 450 (48.1%) Females 485 (51.9%)	US	DNS score
Hu, 2021	China	Cross-sectional	To examine whether serum uric acid in T2DM is influenced by age, gender, BMI, lipid, renal function and other characteristics	2809 T2DM	Males 1784 (63.5%) Females 1025 (36.5%)	US (+reduced alcohol intake)	Physical examination and NCS
Kim, 2014	Korea	Cross-sectional	To assess association between NAFLD and macro- and micro-vascular complications	929 T2DM	Males 489 (52.6%) Females 440 (47.4%)	US	Physical examination and NCS
Leite, 2021	Brazil	Cross-sectional	To evaluate the NAFLD fibrosis score as predictors of complications development and mortality	554 T2DM	Males 218 (39.4%) Females 336 (60.6%)	US + NFS	Physical examination
Lombardi, 2020	Italy	Cross-sectional	To evaluate whether FibroScan^®^ is able to detect an association between hepatic steatosis and micro- and macro-vascular complications	394 T2DM > 5 years	Males 210 (53.4%) Females 184 (46.6%)	US +NFS +FibroScan	Physical examination and NCS
Lv, 2013	Cina	Cross-sectional	To determine the prevalence and risk factors for NAFLD and evaluated its correlations with microvascular complications	1217 T2DM	Males 460 (37.8%) Females 757 (62.2%)	US (+absence of a secondary cause of steatosis)	Physical examination
Mantovani, 2017	Italy	Cross-sectional	To assess association between NAFLD and DPN	286 T1DM	Males 121 (42.3%) Females 165 (57.7%)	US	MNSI score and VPT
Mikolasevic 2021	Croatia	Cross-sectional	To examine whether NAFLD is associated with chronic vascular complications of T2DM	442 T2DM	Males 209 (47.3%) Females 233 (52.7%)	FibroScan	Physical examination and NCS
Tripolino, 2019	Italy	Cross-sectional	To evaluate association between NAFLD and complications	124 T1DM	Males 68 (60.7%) Females 44 (39.3%)	HSI	Physical examination and NCS
Vendhan, 2014	India	Cross-sectional	To estimate the prevalence and clinical profile of NAFLD	736 T1DM	Males 384 (52%) Females 354 (48%)	US	VPT
Williams, 2015	Australia	Cross-sectional	To examine the association between distal VPT and NAFLD	456 T2DM	Males 270 (59.2%) Females 186 (40.8%)	US	VPT
Yan, 2016	China	Cross-sectional	To explore differences in complications when NAFLD developed with pre-existing T2DM	212 T2DM	Males 120 (56.6%) Females 92 (43.4%)	US	Physical examination
Huang, 2021	China	Cross-sectional	To evaluate the relationship between NAFLD and DPN	520 T2DM	Males 227 (43.7%) Females 293 (56.3%)	FibroScan	Physical examination and NCS

BMI, Body Mass Index; DM, Diabetes Mellitus; DNS, Diabetic Neuropathy Symptom; HSI, Hepatic Steatosis Index; MNSI, Michigan Neuropathy Screening Instrument; NAFLD: nonalcoholic fatty liver disease; NCS, Nerve Conduction Study; NFS, NAFLD fibrosis score; T1DM, Type 1 Diabetes Mellitus; T2DM, Type 2 Diabetes Mellitus; US, Ultrasound; VPT, Vibratory Perception Threshold.

## Data Availability

Derived data supporting the findings of this study are available from the corresponding author on request.

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
