# Peer review of "Association of Nonalcoholic Fatty Liver Disease (NAFLD) with Peripheral Diabetic Polyneuropathy: A Systematic Review and Meta-Analysis"

_jcm, 2021, doi:10.3390/jcm10194466_

Round 1

Reviewer 1 Report

This meta analysis contributes significantly in better understandment of NAFLD in association with metabolic comorbidities. Patients with NAFLD and T2DM have significant higher prevalence of diabetic polinuropathy . The combination of rising prevalence and significant potential complication warrant further research into NAFLD and other metabolic comorbidities.

The artical is original, The hypotesis is suported by the results of meta analysis., and the results of the study allow further research to be continued.

Author Response

Response to Reviewer 1 Comments

Point 1

This meta-analysis contributes significantly in better understandment of NAFLD in association with metabolic comorbidities. Patients with NAFLD and T2DM have significant higher prevalence of diabetic polineuropathy. The combination of rising prevalence and significant potential complication warrant further research into NAFLD and other metabolic comorbidities.

Response 1

Thanks for the comment. We observed a significantly increased DPN prevalence among diabetic patients with NAFLD, in particular in case of T2DM. Interestingly, large-scale prospective studies are required to elucidate causal associations between NAFLD and DPN.

Point 2.

The article is original, The hypotesis is suported by the results of meta analysis, and the results of the study allow further research to be continued.

Response 2

Thanks for the comment.

Reviewer 2 Report

This is a very well conducted meta-analysis, presenting original data on the association between NAFLD and DPN.  Before the decision on its publication is taken, the authors should nevertheless address the following critical remarks:

  1. In 90% of cases NAFLD is hardly a separate clinical entity as it is rather a direct complication of overweight or obesity. This relationship (being largely a confounding factor) should be clearly explained in the Discussion section and related to the risk of DPN.
  2.  NCS is not needed to diagnose DPN (as written in second para, page 14) and it hardly ever is used for the diagnosis confirmation.
  3. The word 'parameter' should be replaced with 'variable'

Author Response

Response to Reviewer 2 Comments

This is a very well conducted meta-analysis, presenting original data on the association between NAFLD and DPN.  Before the decision on its publication is taken, the authors should nevertheless address the following critical remarks:

Point 1.

In 90% of cases NAFLD is hardly a separate clinical entity as it is rather a direct complication of overweight or obesity. This relationship (being largely a confounding factor) should be clearly explained in the Discussion section and related to the risk of DPN.

Response 1

Thanks for the comment. The presence of obesity as correlate of PND and NAFLD represents a confounding factor in the evaluation of association NAFLD-PND, at least in DM2. As suggested, we include the following sentence in the Discussion section.

“In details, NAFLD determines a complex array of metabolic and extra-hepatic consequences, which result from the intra-hepatic deposition of ectopic fat. This condition strongly correlates with abdominal obesity, insulin resistance and all components of metabolic syndrome. Notably, obesity is one of the clinical correlates of PND in DM2 people. Therefore, obesity itself could represent a confounding factor of the association between NAFLD and PND, at least in DM2”.

Point 2

NCS is not needed to diagnose DPN (as written in second para, page 14) and it hardly ever is used for the diagnosis confirmation.

Response 2

Thanks for the comment. NCS is not necessary for the clinical diagnosis of PND and is certainly used very little in clinical practice. We agree with the comment. Considering the number of studies that reported NCS, for the present meta-analysis the NCS data was used as sensitivity analyses in order to reduce the high heterogeneity among studies. We used the terms probable (signs and symptoms) and confirmed (NCS) PND in order to use the official classification (Toronto Consensus).

Point 3

The word 'parameter' should be replaced with 'variable'

Response 3

Thanks for the suggestion and edit the text using the term “variable”.
